# AC Electric-Field Assistant Architecting Ordered Network of Ni@PS Microspheres in Epoxy Resin to Enhance Conductivity

**DOI:** 10.3390/polym13213826

**Published:** 2021-11-05

**Authors:** Zhiliang Han, Jinlu Wang, Qingliang You, Xueqing Liu, Biao Xiao, Zhihong Liu, Jiyan Liu, Yuwei Chen

**Affiliations:** 1Key Laboratory of Optoelectronic Chemical Materials and Devices, Ministry of Education, Jianghan University, Wuhan 430056, China; silence__han@163.com (Z.H.); biaoxiao@jhun.edu.cn (B.X.); liuzh@jhun.edu.cn (Z.L.); 2Flexible Display Materials and Technology Co-Innovation Center of Hubei Province, Jianghan University, Wuhan 430056, China; 18271908753@163.com (J.W.); yql1976@126.com (Q.Y.); 3Key Laboratory of Rubber-Plastics, Ministry of Education, Qingdao University of Science and Technology, Qingdao 266042, China

**Keywords:** nickel-coated polystyrene microspheres, electric fields, polymer–matrix composites, conductivity, electrical properties

## Abstract

By using the low loading of the conductor filler to achieve high conductivity is a challenge associated with electrically conductive adhesion. In this study, we show an assembling of nickel-coated polystyrene (Ni@PS) microspheres into 3-dimensional network within the epoxy resin with the assistance of an electric field. The morphology evolution of the microspheres was observed with optical microscopy and scanning electron microscopy (SEM). The response speed of Ni@PS microsphere to the electric field were investigated by measuring the viscosity and shear stress variation of the suspension at a low shear rate with an electrorheological instrument. The SEM results revealed that the Ni@PS microspheres aligned into a pearl-alike structure. The AC impedance spectroscopy confirmed that the conductivity of this pearl-alike alignment was significantly enhanced when compared to the pristine one. The maximum enhancement in conductivity is achieved at 15 wt. % of Ni@PS microspheres with the aligned composites about 3 orders of magnitude as much as unaligned one, typically from ~10^−5^ S/m to ~10^−2^ S/m.

## 1. Introduction

Electrically conductive adhesives have widely applied in electronic devices such as die attachment, solderless interconnection, semiconductor packaging [1,2], interconnection of solar cells [3,4] and touch screens [5]. The conductive adhesives composed of polymer matrices in which conductive fillers are dispersed. The widely used polymer matrix is epoxy resins owing to their high chemical and thermal resistance, excellent mechanical properties, good adhesion to various materials, availability of solvent free formulations, ease of controlling viscosity [6,7]. The conductive fillers include metal fillers (silver, gold, nickel, copper) [8,9], carbon-based fillers (carbon nanotube, carbon black, carbon nanofibers, graphene etc.) [10,11,12], ceramic fillers [13,14], and metal-coated fillers [15,16,17]. In the conductive adhesives, the filler creates a conductive network above its percolation threshold and thus provides electron conductivity paths. For nano or micron-sized metal fillers such as silver or nickel embedded polymer, the required percolation threshold of the filler loading is up to 70–80% by weight or 23–30% by volume fraction [18,19]. The much large fraction of precious metal is a great cost, and same time causes a negative effect on adhesion properties such as brittleness, thermal expansion mismatch between the components. In addition, the density of silver and nickel is about 8–10 times of the epoxy resin, the metal will deposit on the bottom of the adhesive during the epoxy curing.

Compared to the pure metal filler, core-shell conducting fillers are gotten more and more attention due to lower cost and reduced density. The core is usually consisted of polymer such as polystyrene and polymethyl methacrylate microspheres with diameter in nano to micron-meter sizes, whereas the shell is covered by conductive metals with thickness in a few hundred nanometers. Gold and silver layers are excellent conductivity, but gold is high cost. silver layer has some limitations in their end applications such as Ag-migration, poor interface adhesion with the polymer matrix and easy to form Ag_2_O in the air atmosphere, whereas nickel with outstanding combination property such as its high chemical stability, lower density than gold and silver, and easy to be coated on the surface of polymer core by electroless plating, is continuously to be of great interest as a conductive filler in epoxy resin.

To achieve a high electric conductivity of composites at a conductive filler loading as low as possible, applying magnetic field [20,21,22,23] or electric field [24,25] to assemble particles into ordered structure have proven to be an efficient approach. Such an assembly might produce the continuous conductive pathway at a lower filler loading. Johnsen aligned silver micron powder in epoxy through electric field to enhance the electric conductivity [26]. Sanctuary investigated the alignment of nickel particles in the epoxy matrix by magnetic field, forming adhesive with anisotropic conductivity [27]. In addition, carbon nanotubes [28], carbon nano-cones [29] and metallic micro-coils [30] were aligned into microscopic wires under an alternating-current electric-field, forming one-dimension conductive pathways with conductivity enhancement of at least 2–3 orders of magnitude. In our previous work, electric-field was applied to induce alignment of ceramic nanoparticles to architect three-dimension pathways in poly (dimethyl siloxane) to enhance the ion conductivity [31].

In current study, the Ni coating PS microspheres (Ni@PS) were prepared and used as conductive filler of epoxy resin. With the assistant of the electric field, the ordered and continuous pathway composed of the Ni@PS microspheres was built-up. Evolution of the morphology of Ni@PS microspheres in epoxy was in-situ observed, the response speed of the Ni@PS microspheres to the electric field was characterized by measuring the viscosity and shear stress variation of the suspension under electric field at low shear rate. The enhancement of electric conductivity of aligned composites was demonstrated with the AC impedance spectroscopy.

## 2. Materials and Experiments

### 2.1. Materials

Polystyrene (PS microspheres with particle an average diameter of 3.8 μm were purchased from Shanghai Chemical Reagent Co. (Shanghai, China). Chromium trioxide (CrO_3_), H_2_SO_4_ (98%), tin (II) chloride dehydrate (SnCl_2_·2H_2_O), aqua ammonia (NH_3_·H_2_O), HCl, Hydrazine hydrate (N_2_H_4_·H_2_O), absolute ethanol (95%) was supplied by Sinopharm Chemical Reagent Co. Ltd. (Shanghai, China) and used as received. Tin chloride (SnCl_2_·2H_2_O), sodium stannate (NaSnO_3_·3H_2_O), palladium chloride (PdCl_2_). Nickel sulfate (NiSO_4_·6H_2_O), sodium hypophosphite (NaH_2_PO_4_·H2O, 98%). Sodium pyrophosphate dec-hydrate (Na_4_P_2_O_7_·10H_2_O, 98%), triethanolamine (99.9%), ethylene diamine (99.9%), thiourea (99.9%), ammonia, hydrochloric acid (HCl, 36% by weight, and ethanol (95%). All these chemicals were purchased from Xi’an Chemical Regent company (Xi’an, China) and used as received. Epoxy resin (EP, Diglycidylether of bisphenol-A with an average epoxide equivalent of 0.51 equivalents/100 g), hardener 2-methyl imidazole (99.9%) and diluter allyl 2,3-epoxypropyl ether (98.0%) were provided by Baling Petrochemical Company (Yueyang, China). Copper conductive tapes, ITO conducting glasses and electric tapes were commercially obtained from 3M Co. Ltd (Shenzhen, China). 

### 2.2. Preparation of Ni@PS Microspheres

Figure 1a shows the preparation procedure of Ni@PS microspheres. Typically, 5 g PS microspheres were immersed in 6M sulfuric acid for 60 min at 30 °C, filtered and washed with deionized water until no sulfate remained. The roughened PS microspheres were dipped in an acid SnCl_2_ solution (20 g/L SnCl_2_ and 20 mL/L HCl (37%)) for 60 min at 50 °C and then rinsed with deionized water until no Cl−remained. The deposition reaction of the electroless nickel occurs at palladium catalytic active centers that absorbed on surface of the PS. The activation was carried out in an acid PdCl_2_ solution composed of SnCl_2_ (0.343 M/3.414 M HCl), Na_2_SnO_3_ (0.026 M M/3.414 M HCl), PdCl_2_ (0.006 M/3.414 MHCl) for 60 min at 40 °C. The activated PS microspheres were separated, rinsed with deionized water, and dried at 60 °C in vacuum [32,33].

Nickel plating on PS Microspheres: The activated PS microspheres (0.2 g) were immersed in 100 mL of a nickel-plating solution consisting of NiSO_4_·6H_2_O (40 g/L), NaH_2_PO_2_·H_2_O (30 g/L), Na_4_P_2_O_7_·10 H_2_O (80 g/L), CH_4_N_2_S (1.5 mg/L) and (CH_2_CH_2_OH)_3_N (100 g/L) with mechanical stirring at 75 °C for 90 min. The gray-black Ni@PS microspheres were filtered and rinsed with distilled water, dried at 60 °C.

### 2.3. Preparation of Ni@PS/EP Composites

The Ni@PS/EP composites were fabricated using the self-made setup as shown in Figure 1b. Ni@PS microspheres were dispersed in the matrix containing EP oligomer, dilute and curing agent using Hielscher UP400S ultrasonic processor and planetary centrifugal mixer (Thinky Mixer ARE-310, Tokyo, Japan). The weight ratio of EP oligomer, dilute and curing agent is 100:10:25. The Ni@PS content in the composite is based on the total weight of the matrix. Then suspension was put between two ITO coated glasses. The ITO surface was coated with silicone release tape. The one electrode was fixed onto a hot plate. The oriented structure is obtained by applying AC electric field of 300 V/mm of the for 10 min at 80 °C. The sample was post-cured at 80 °C for another 120 min after removing the electric field. The isotropic Ni@PS/EP and blank epoxy are prepared using the same curing cycle without electric field.

### 2.4. Measurement and Characterization

The infrared spectroscopy was conducted on a Fourier Transform infrared spectrometer Tensor 27 (Bruker, Germany). The thermogravimetric analysis (TG) was performed in N_2_ atmosphere at a heating rate of 20 °C/min in the temperature range of 30–700 °C with thermogravimetric analyzer (Netzsch TG 209 F1, Selb, Germany). The morphology of original PS, Ni@PS microspheres and cross-section of cured Ni@PS/EP composites were investigated by scanning electron microscopy (SEM) (Hitachi SU8010, Tokyo, Japan). The cured Ni@PS/EP composites were broken at ambient temperature of about 25 °C and coated with Au. The generator was operated at 40 kV and 40 mA with a beam monochromatized to Cu Kα radiation. The chemical compositions of the microspheres were analyzed by energy dispersion X-ray analysis (EDS) instrument (IXRF System 550i, Austin, TX, USA). X-ray diffraction (XRD) with Cu Kα radiation was used to analyze the crystal texture of Ni-coating on a XRD instrument (Rigalcu D/max C, Tokyo, Japan).

The alignment of Ni@PS microspheres in EP mixture under AC electric filed was in-situ monitored by optical microscope (Zeiss Axiolab 5, Oberkochen, Germany) equipped with digital camera at room temperature. The suspension was placed onto a glass slide. Two parallel copper tapes were used as the electrodes connecting with the AC electric field. The in-situ images were taken with a DC 290 Kodak Zoom digital camera connected to the optical microscope.

Electrorheology experiments were carried out with rheometer (TA ARES-G2, New Castle, DE, USA) equipped with an EP cell having parallel plates with a diameter of 40 mm. The gap between the parallel plates which served as electrodes, was kept constant at 0.5 mm. The device was connected to the AC electric field-generating apparatus. Transient response of the Ni@PS microspheres to an external AC field in EP suspension is recorded by performing time-sweeps at a shear rate of 0.01 s at 30 °C.

The electrical conductivity was measured after the samples curing, locking the aligned chain in a conducting configure ration. The measurement was performed on AC electric impedance analyzer (Beiguang GDAT-S, Beijing, China) with applied frequencies ranging from 100 Hz to 1 MHz.

## 3. Results and Discussion

### 3.1. Characterization of Ni@PS Microspheres

Figure 2 presents the morphology and composition of the original PS and Ni@PS microspheres. The original PS microspheres have a smooth surface and uniform size with diameter about 3.8 μm. The EDS demonstrate that PS microspheres have 51.645 wt.% of carbon and 32.525 wt.% of oxygen. Au of 15.562 wt.% is detected due to spraying of pretreatment. The surface of Ni@PS microspheres is rough and composed of tiny solid grains. The element detected from EDS is as following: 86.194 wt.% nickel, 10.086 wt.% phosphorus. 0.192 wt% carbon, 1.525 wt.% of oxygen. The phosphorus is from the nickel-plating process. Carbon and oxygen are attributed to the PS substrate because the depth of the electron beam ripping into the coating is about 1 μm.

Figure 3a shows the thermal decomposition of the PS and Ni@PS microspheres. The pure PS starts to decompose at 398 °C and almost loses all the weight before 450 °C. The Ni@PS microspheres is more thermal stable than the PS and the initial decomposition temperature moves to 450 °C with the residues of 50.4% obtained at 700 °C. The reason is that the Ni acts as a protecting layer to avoid the heat transferring from the atmosphere to the bulk PS. In XRD spectrum of the microspheres (Figure 3b), the peaks for PS microspheres locate at 2θ = 10° and 20°, respectively. In the XRD of Ni@PS microspheres, a peak around 44.40°appearsthat typical diffractive spectrum of pure nickel (PDF No. 04-0850), indicating that Ni has wrapped the whole surface of the PS microspheres. Figure 3c presents the FTIR spectra of original PS and Ni@PS microspheres. In the original PS, the absorption peaks at 1450–1600 cm^−1^ are due to the aromatic ring and the peaks at 700–755 cm^−1^ are from vibration of -CH_2_ of PS, while in the FTIR of Ni@PS microspheres, the absorptions for PS have disappeared, indicating the PS has been covered by the Ni.

### 3.2. Alignment of Ni@PS in EP Suspension under AC Electric Field

The suspended Ni@PS microspheres in the EP matrix will be polarized in the electric field and result in electric dipoles. The dipoles will interact with each other to form the chains. The metallic particles are nearly infinitely polarizable, thus, Ni@PS microspheres can align at lower electric field strength in a viscous polymer matrix than non-conductive particles do, as clay, PS microspheres, or semiconductive ceramic particles. The particle concentration and electric field strength are important factors for alignment. They must exceed a threshold value for the chain forming and growing [34].

Figure 4 presents the optical images of the Ni@PS microspheres after applying different electric field strength for 10 min. The experiment set-up for investigation of alignment via optical microscope is shown in the Figure 4f. The Ni@PS microspheres distribute randomly in the in EP when no electric field is applied. At the 200 V/mm of electric field strength, the chains feature short and irregular. With the electric field strength increasing to 300 V/mm, the chain becomes more regular and chain length increases. At 400 V/mm of electric field strength, the chain slightly distorts and becomes irregular again. At 500 V/mm of electric field strength, the chains branch out in the different direction. In our experiment, the electric field intensity is set at 300 V/mm with the frequency of 1000 Hz for fabricating the composites.

Figure 5 shows the optical morphologies of Ni@PS microspheres at different time under 300 V/mm of electric field strength. In the suspension with 1 wt.% and 3 wt.% of Ni@PS, the microspheres have organized into small chains anchoring to the electrodes or floating in the matrix within 3 min. After 5 min, the length and number of chains are enlarged while the number of isolated microspheres decrease. After applying the electric field for 7 min, chain length on the electrodes don’t change anymore. Meantime, the microspheres, or chains in the matrix become sparse. For the suspension containing 5 wt.% Ni@PS microspheres, the chains have spanned the electrodes after 5 min. Results in Figure 5 indicates that microspheres preferentially anchor to the electrode and growing up into chain. This is because that electrostatic forces between the electrode and microspheres are greater than that of microspheres themselves [35].

### 3.3. Rheology of a Ni@PS Microspheres in Epoxy under High AC Electric Fields

Rheology is a promising method to probe the microstructure of Ni@PS microspheres in EP oligomer under the electric field. At a low shear rate, the applied shear field is only capable to slightly distort (but not destroy) the microstructures of microspheres and the viscosity of the suspension is approximately constant. Once chains of microspheres span the gap between electrodes, a finite stress, the static yield stress, is required to break the chains and subsequently the viscosity of the suspension increases. To obtain the viscosity curve under a shear rate of allowing to retain structural equilibrium. The steady shear flow tests were performed in controlled shear rate at 0.01/sunder electric field strength of 300 V/mm.

Figure 6a shows the change in the shear viscosity (η) with time of the suspension containing different Ni@PS content. In the first 30 s, the suspension shows a Newtonian behavior with the viscosity remaining constant because no voltage is applied. The microspheres exhibit randomly distributed configuration. After 300 V/mm of electric field strength is applied, an abrupt increase in the viscosity is observed which demonstrates the formation of the chain structure. It is known that the viscosity is ratio of shear stress to shear rate. After alignment, the microsphere content within the chain increases dramatically, so the higher yield stress is required, which result in a jump in the viscosity between the aligned and random dispersed microspheres suspension. According to the Walsh and Saar [36] suggestion, yield stress increases as a power growth function of solid microsphere volume fraction for a suspension at a steady shear rate. Therefore, the viscosity variation with Ni@PS microsphere content is non-linear, as shown in Figure 6b.

To evaluated how quick the Ni@PS microspheres to build up into chain in the EP suspension when subjected to electric field. The shear viscosity with time is fitted with a typical function the sigmoidal-Boltzmann expression [37]:(1)$$\eta =\frac{\eta \left(0\right)-\eta \left(E\right)}{1+exp\left(\frac{\left(t-t_{0}\right)}{\left(t_{res}/4\right)}\right)}+\eta \left(E\right)$$
where, $t_{0}$ is the time corresponding to a viscosity value [*η*(0) + *η*(*E*)]/2, and *t_res_* is the response time required by the suspension to change from *η* (0) to its final value *η*(*E*). The response time, $t_{\mathrm{res}}$ demonstrates the speed of the chain buildup, which plotted as functions of the Ni@PS content, as shown in the Figure 6b. the $t_{\mathrm{res}}$ decreases as Ni@PS content increasing. For instance, $t_{\mathrm{res}}$ is 29 min for the suspension containing 5 wt.% Ni@PS, and $t_{\mathrm{res}}$ reduce to 16 s when Ni@PS content increases to 15 wt.%. however, $t_{\mathrm{res}}$ does not change when the Ni@PS content above 25 wt.%. At low Ni@PS content, the polarized microspheres took a longer journey to attract another one, so a longer time is required for completing the construction of the chain. At high Ni@PS content, the distance between microspheres is short, and microspheres easily and quickly attract each other to form the chain, so $t_{\mathrm{res}}$ is low.

### 3.4. Morphology of Ni@PS/EP Composites

The morphology of Ni@PS/EP composite containing 3 wt.% of Ni@PS was characterized by SEM-EDS. The microspheres distribute randomly in film without the application of electric field, as seen in Figure 7a. Short chains are observed in film with the application of 300 V/mm electrical field (Figure 7b). Under the high magnification, it can be noticed that microspheres pile up tightly within the chain. Some of the Ni has peeled off from microspheres, as evidenced by the red circle morphology for the mapping of Ni element in the EDS (Figure 7d).

Figure 8 presents the SEM morphology of the composites with Ni@PS content from 5 wt.% to 20 wt.%. The chains are discontinuous within the films containing 5 wt.% of Ni@PS, as marked in red in the Figure 8a, which is different from the image under optical microscopy that chains are continuous along the electric field direction in the suspension containing 5 wt.% Ni@PS (Figure 5). The reason is that the optical image is a superposition of microspheres of different layer along the thickness direction of the film (electric field direction). SEM image gives the information of the film surface where the only microspheres exposed at the surface can be observed. Therefore, the number of microspheres looks fewer, and chains feature interrupted in the SEM morphology. In the films containing 10 wt.% of Ni@PS, the chains are denser and exhibit a cone structure. The cone structure results from the gravitational force acting on each microsphere (or microsphere clusters) during electric-field induced alignment. The small microspheres can overcome the gravitational pull, while the larger microsphere clusters are too heavy and migrate toward the bottom of the suspension during the dielectrophoretic process and created the base portion of the cone. In the film containing 15 wt.% and 20 wt.% of Ni@PS the chains are distorted, and chains crosslinked slightly forming the 3D network. Comparing with the aligned composites, the microspheres disperse disorderly and isolated each other in the unaligned ones (Figure 8a’–d’).

As the Ni@PS content increasing to 25 wt.%, the chains have linked with each other. In the composites containing 25 wt.% Ni@PS, microspheres arrange more regular in the network with application of the electric field than that without electric field (Figure 9a,a’). In the film containing 30 wt.% of Ni@PS, the anisotropic chain structures can be barely seen (Figure 9b,b’). The microspheres with or without applying the electric field show similar network configuration within the film. The reason is that at such high microspheres content and high spatial density, the movement of microspheres is frustrated in the viscous matrix and alignment process was interrupted.

### 3.5. Electrical Conductivity of Ni@PS/EP Composites

The electrical conductivity of cured Ni@PS/EP composites with different Ni@PS content was measured along the thickness direction. Log–log plots of the AC conductivity of neat epoxy and Ni@PS/EP composites as a function of frequency at room temperature are shown in Figure 10a,b. The conductivity is both frequency and Ni@PS content dependent. The conductivity of neat epoxy and the samples with low Ni@PS content present curves similarities. For example, the unaligned samples with 5–15 wt.% Ni@PS or the aligned sample with 5 wt.% Ni@PS, the conductivity tends to be constant values in frequency range about 20–1000 Hz, then increases in the frequency between 1000–0.1 M Hz. Moreover, as the Ni@PS content increases, the onset of the frequency dependent area shifts to higher value. When the Ni@PS content is above 20 wt.% for the unaligned samples and above 10 wt.% for the aligned samples, the conductivity spectra exhibit plateau in the whole range of applied frequency.

In the Figure 10, plateau of nearly frequency-independent conductivity is referred to as the DC conductivity. At a given temperature, the response of conductivity with frequency could be expressed by the empirical power law, that first described by Jonscher [38], as shown below:$$\sigma_{AC}\left(\omega \right)=\sigma_{DC}+A{\left(2\pi f\right)}^{n}$$
where *f* is the frequency applied, *A* and *n* are parameters dependent temperature, morphology, movement of polymer chains, and composition. For the nanotubes filled epoxy composites [39], the critical frequency (f*_c_*) value that switching DC conductivity ($\sigma_{DC}$) to AC conductivity $\left(\sigma_{AC}\right)$ dominant increases with the filler content. It was found that f*_c_* increases from 10 Hz to 1.45 MHz as the nanotube content increases from 0.1 to 1.0 wt%. In our study, there is no frequency-independent conductivity appearing in the spectra for the samples with high Ni@PS content because the f*_c_* value is beyond the permissible limit of applied frequency 1 MHz [40].

It can be noticed from Figure 10a,b that the conductivity of all the samples at lower frequency (<1000 Hz) is frequency-independence. To investigate the effect of alignment of Ni@PS on the conductivity without considering the contribution of the frequency. The values of electrical conductivity of the samples with different Ni@PS content obtained at frequency of 1000 Hz was plotted, as shown in Figure 10c.

The conductivity of pure EP and Ni@PS microspheres is 1.4 × 10^−7^ S/m and 1 S/m, respectively. The conductivity increases with Ni@PS content for both aligned and nonaligned composites. The aligned composites (red curve) exhibit higher electrical conductivity than nonaligned ones (black curve) at all filler loading. For instant, the aligned composite containing 5 wt.% Ni@PS has a same electrical conductivity level as that of the nonaligned composite containing 15 wt.% Ni@PS. The conductivity of aligned composite is 1000 times of the unaligned one when the Ni@PS content is at 15 wt.%.

The increase in conductivity stems from the microsphere aggregation and emerging conductive pathways. There should be a percolation content of the conductive fillers and a significant enhancement in the electrical conductivity could appear around this doping amount based on the formation of a 3D network of the conductive fillers. In our experiment, a conductivity percolation threshold ($\varphi_{c}$) of the Ni@PS is evaluated by fitting the experimental data to a Kirkpatrick’s equation [41,42,43,44]:$$\sigma \;\approx \;\sigma_{0}{\left(\varphi -\;\varphi_{c}\;\right)}^{p}$$
where σ is conductivity of the samples, $\sigma_{0}$ and p are constants, φ is the filler concentration by volume. Figure 10d,e is log σ plotted against log ($\varphi -\varphi_{c}$) for the Ni@PS/EP composites with randomly dispersed and aligned Ni@PS, respectively. The conductivity percolation threshold is estimated as 9.72 vol% (11.92 wt.%) for aligned composites and 15.63 vol% (19.14 wt.%) for unaligned one. The $\varphi_{c}$ of the aligned composites is reduced by 37.72% based on the mass fraction, compared to unaligned composites with randomly dispersed Ni@PS. Thus, these results clearly demonstrate that applying the electric field could significantly reduce the content of Ni@PS necessary to achieve percolation.

In the study by Untereker [45], the conductivity of various particle filled polymers never exceeded 1% of that of the pure particle material itself when filler loading at percolation threshold value. The conductivity of Ni@PS microspheres is about 1 S/m. The maximum conductivity for both the aligned and unaligned composite is about 10^−2^ S/m, which is agree with the conclusion from Untereker’s report.

In addition, the conductivity gap between the aligned and unaligned composite becomes small at Ni@PS content above 25 wt.%. The reason is that at high Ni@PS content, spatial density of microspheres became too high leading to interruption of alignment.

## 4. Conclusions

In summary, Nickel coated PS (Ni@PS) conductivity microspheres have been successively prepared and uses as the conductor filler for the epoxy adhesive. By applying alternating electric field, Ni@PS microspheres assemble and align into continuous conductive pathway in epoxy. In-situ optical microscopy observation indicate that microspheres chains grow up at electrodes and extend to the opposite electrode by attracting the isolated microspheres. Rheology test showed that t_res_ decrease with Ni@PS content increasing to 20 wt.%. The t_res_ does not change after Ni@PS content beyond 25 wt.%. The SEM morphology shows that well-organized 3D network was formed at 15 wt.% of Ni@PS under 300 V/mm of the electric field strength. However, the alignment of microspheres is stunted as the Ni@PS content reaching to 30 wt.% due to a lack of free space. The aligned composites have an enhanced electric conductivity with conductivity 1000 times of unaligned one at 15 wt.% of Ni@PS. Moreover, a higher conductivity level may be obtained by optimizing thickness of Ni layer, adopting different metal, and changing interfacial properties between conductive microspheres, and between microspheres and the electrodes.

## Figures and Tables

**Figure 1 polymers-13-03826-f001:**
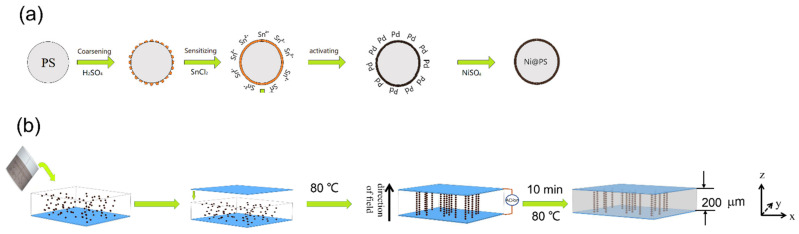
Schematics of preparing of Ni@PS microspheres (**a**) and aligned Ni@PS/EP composites (**b**).

**Figure 2 polymers-13-03826-f002:**
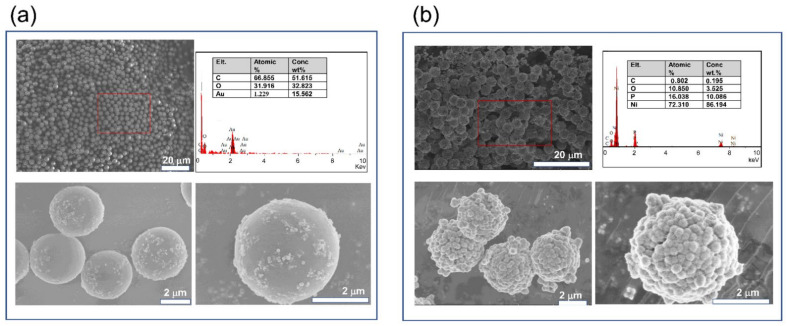
Surface morphology of (**a**) original PS, and (**b**) Ni@PS microspheres by SEM.

**Figure 3 polymers-13-03826-f003:**
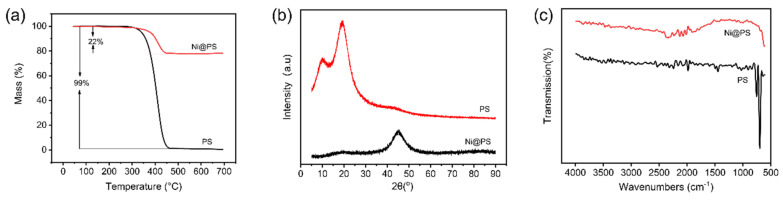
(**a**) TG curves, (**b**) XRD spectra, and (**c**) FTIR spectra of original PS and Ni@PS microspheres.

**Figure 4 polymers-13-03826-f004:**
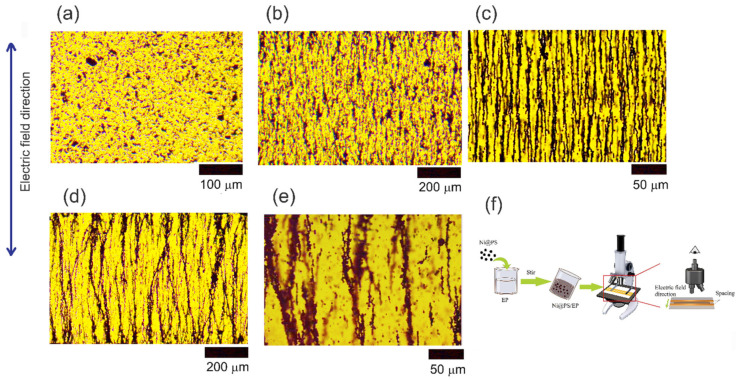
Optical microscopy of Ni@PS microspheres in EP oligomer under different electric field strength (V/mm): (**a**) 0, (**b**) 200, (**c**) 300, (**d**) 400, (**e**) 500 (frequency 1000 Hz, aligning time10 min, temperature 30 °C). (**f**) The set-up for monitoring alignment of Ni@PS microspheres in EP with optical microscope.

**Figure 5 polymers-13-03826-f005:**
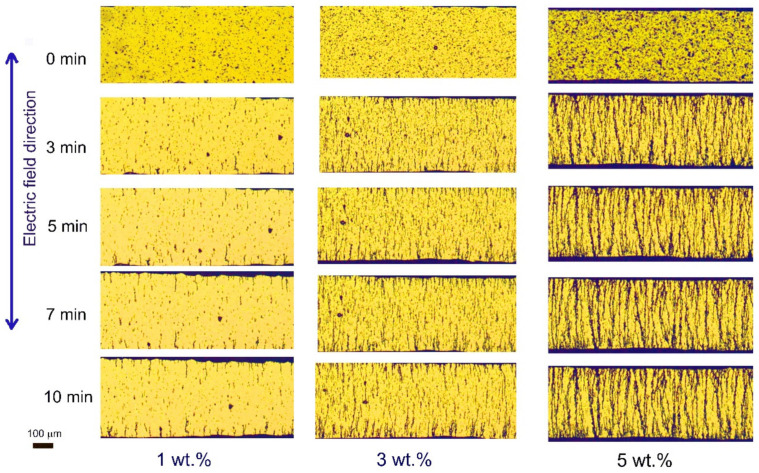
Effects of the concentration of the Ni@PS microspheres on the alignment.

**Figure 6 polymers-13-03826-f006:**
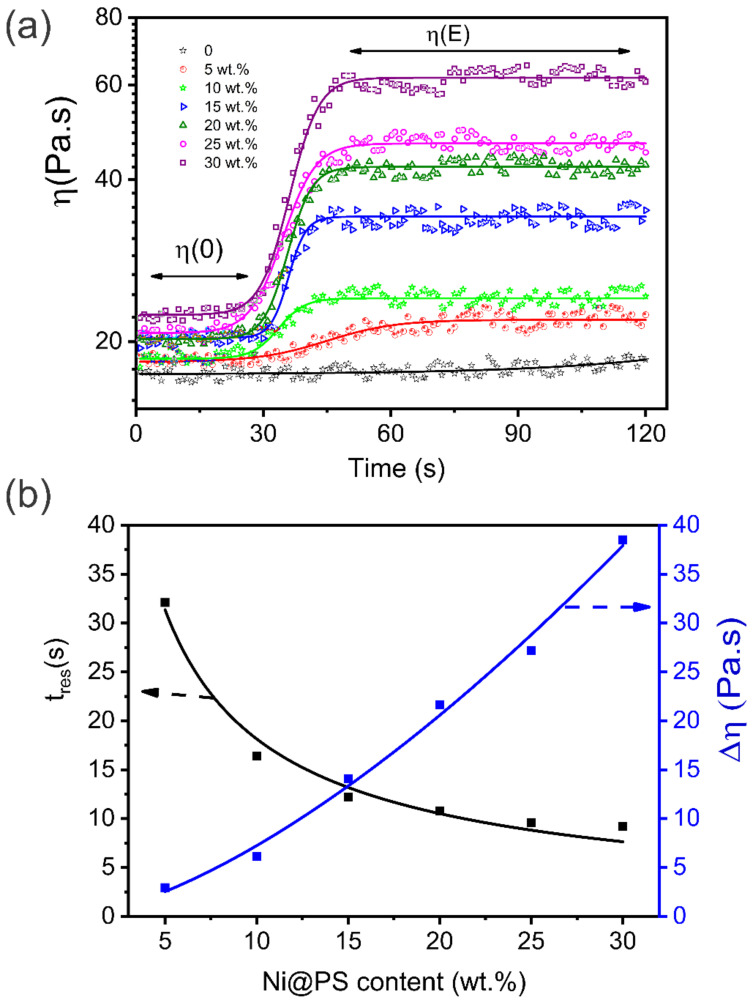
(**a**) Plots of the shear viscosity vs. time at constant shear rate, as a function time under different electric field strength. The field is applied at t = 30 s. Lines correspond to the best fit to a Sigmoidal-Boltzmann curve. (**b**) Viscosity increases $\left(\Delta \eta \right)$ and response time $\left(t_{res}\right)$ vs. Ni@PS content.

**Figure 7 polymers-13-03826-f007:**
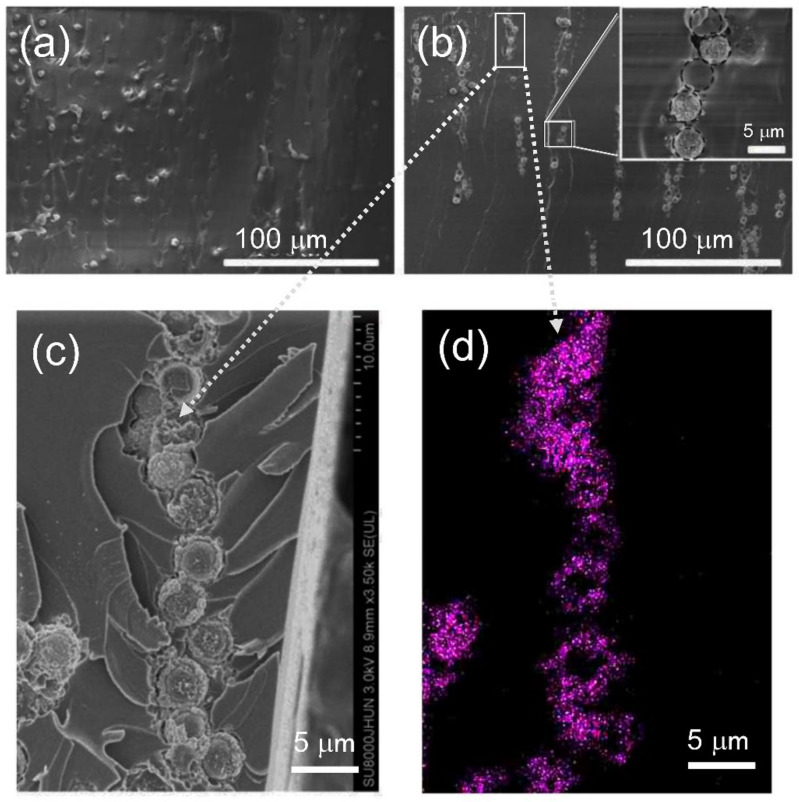
Cross-sectional SEM morphology of 3 wt.% Ni@PS/EP composites (**a**) unaligned, (**b**,**c**) aligned under 300 V/mm electric field strength, (**d**) EDS mapping of Ni corresponding to the graph.

**Figure 8 polymers-13-03826-f008:**
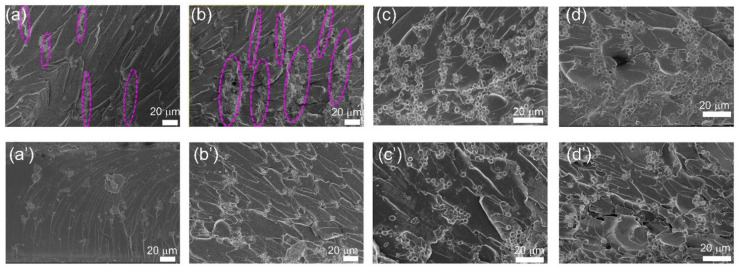
Cross-sectional morphology of Ni@PS/EP composites obtained with 300 V/mm of electric field strength at different Ni@PS content (wt.%): (**a**) 5, (**b**) 10, (**c**) 15, (**d**) 20. (**a’**–**d’**) Corresponding to the composites containing 5 wt.%, 10 wt.%, 15 wt.%, and 20 wt.% of Ni@PS without application of electric field.

**Figure 9 polymers-13-03826-f009:**
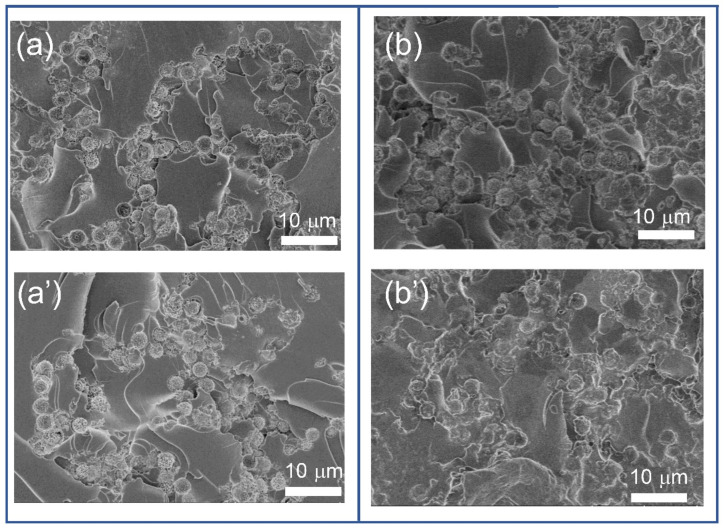
Cross-sectional morphology of Ni@PS/EP composite containing 25 wt.% (**a**,**a’**) and 30 wt.% (**b**,**b’**) of Ni@PS microspheres (**a**,**b**) with 300 V/mm of electric field strength, (**a’**,**b’**) without application of electric field.

**Figure 10 polymers-13-03826-f010:**
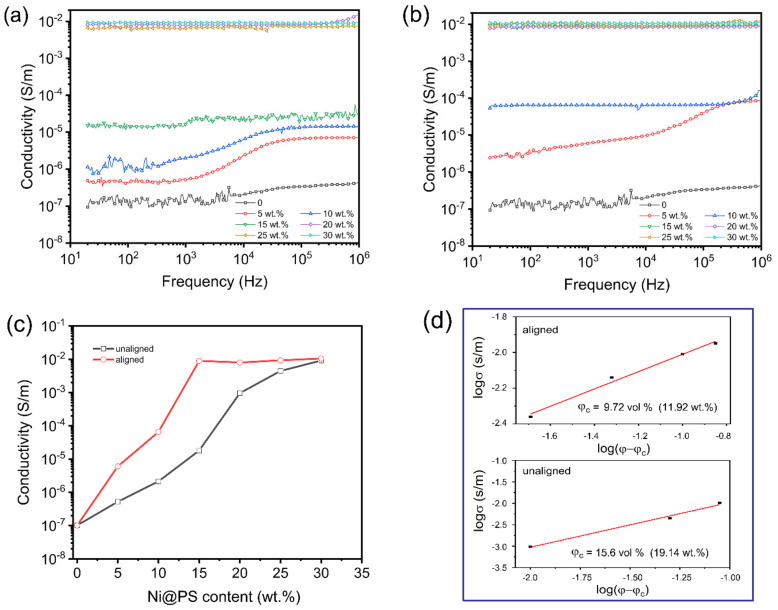
Dependence of electrical conductivity of Ni@PS/EP composites on the frequency at room temperature: (**a**)without application of electric field, (**b**) with 300 V/mm of electric field strength. (**c**) Electrical conductivity of Ni@PS/EP composites with different Ni@PS content at 1000 Hz. (**d**) log σ plotted against log ($\varphi -\varphi_{c}$) for the Ni@PS/EP composites with unaligned and aligned Ni@PS, respectively.

## Data Availability

Not applicable.

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
