# Peer review of "AC Electric-Field Assistant Architecting Ordered Network of Ni@PS Microspheres in Epoxy Resin to Enhance Conductivity"

_polymers, 2021, doi:10.3390/polym13213826_

Round 1
Reviewer 1 Report
The article deals with the development and characterization of composites made of an ordered network of Ni@PS microspheres in epoxy resin. The work is of good technical and scientific quality. The paper is well organized and comprehensive, it presents relevant and numerous experimental results with appropriate interpretations, excepted for the electrical conductivity part p.11.
The following issues should be appropriately addressed before acceptance for publication.
P2 line 54, not complete sentence and no associated reference.
The author should give more details on the interest to use such conductive fillers? What are the advantages over Nickel for example?
Lines 119-120 are not in a good location. As they are related to XRD instrument, they should be placed before the sentence line 117.
Fig 2: The captions including the size are very difficult to read, the author should correct this point.
The Fig 4 (f) is not very useful.
The discussion on conductivity results and mechanisms (p11) should be rewritten, due to misintepretation concerning DC conductivity and percolation threshold.
The sentence “There are two conductive mechanisms existing in the microsphere filled polymer: DC conductive (Ni@PS) and capacitive (polymer)conductive” is not correct. Tha author shoud refer to the to the universal law of Jonsher. The DC conductivity is the independent frequency conductivity at low values of frequency. And the DC conductivity occurs in polymer such as epoxy resin.
Line 294 change “captative” by “capacitive”.
The value given for the percolation threshold value are not correct. The author should use the percolation law to determine the value.
To go deeper in the interpretation, other dielectric measurements at low frequency (0.1 Hz) and as a function of the temperature would be helpful.
Author Response
Response to Reviewer 1 Comments
The article deals with the development and characterization of composites made of an ordered network of Ni@PS microspheres in epoxy resin. The work is of good technical and scientific quality. The paper is well organized and comprehensive, it presents relevant and numerous experimental results with appropriate interpretations, excepted for the electrical conductivity part p.11.
The following issues should be appropriately addressed before acceptance for publication.
Point 1.P2 line 54, not complete sentence and no associated reference.
Response 1: Corrected accordingly. Please see the revised P2 line 66: In our previous work, electric field was applied to induce alignment of ceramic nanoparticles….
Associated reference is as following:
- Liu, X.; Peng, S.; Gao, S.; Cao, Y.; You Q.; Zou, L.; Liu, Z.;, Liu, J.Electric-field directed parallel alignment architecting 3d lithium ion pathways within solid composite electrolyte. Acs Appl Mater Interfaces. 2018, 10, 15691-15696. https://doi.org/10.1021/acsami.8b01631
Point 2: The author should give more details on the interest to use such conductive fillers? What are the advantages over Nickel for example?
Response 2:Corrected accordingly. We have added the more information about the advantages of Ni coated PS in the revised introduction part (P2/14, line 44-54, as shown in the following: Compared to the pure metal filler, core-shell conducting fillers are gotten more and more attention due to lower cost and reduced density. The core is usually consisted of polymer such as polystyrene and polymethyl methacrylate microspheres with diameter in nano to micron-meter sizes, whereas the shell is covered by conductive metals with thick-ness in a few hundred nanometers. Gold and silver layers are excellent conductivity, but gold is high cost. silver layer has some limitations in their end applications like Ag-migration, poor interface adhesion with the polymer matrix and easy to form Ag2O in the air atmosphere, whereas nickel with outstanding combination property such as its high chemical stability, lower density than gold and silver, and easy to be coated on the surface of polymer core by electroless plating, is continuously to be of great interest as a conductive filler in epoxy resin.
Point 3: Lines 119-120 are not in a good location. As they are related to XRD instrument, they should be placed before the sentence line 117.
Response 3: Line 119-120 has been relocated before XRD section. To make the SEM analysis understood better, the sentence has corrected as following: The morphology of original PS, Ni@PS microspheres and cross-section of cured Ni@PS /EP composites were investigated by scanning electron microscopy (SEM) (Hitachi SU8010, Japan). The cured Ni@PS /EP ….
Point 4: Fig 2: The captions including the size are very difficult to read, the author should correct this point.
Response 4:Fig 2 has been replotted. We hope the new one can be clearer.
Point 5: The Fig 4 (f) is not very useful.
Response 5: The comment is good. However, there will have a blank area in the Figure 4 after Fig4 (f) was deleted. Considering the figure 4 composed of 6 subgraphs looks symmetry and full. We hope to keep Fig 4 (f). Moreover, the author might understand better about the particle alignment if Figure 4f was kept.
Point 6: The discussion on conductivity results and mechanisms (p11) should be rewritten, due to mis interpretation concerning DC conductivity and percolation threshold.
Response 6: Corrected accordingly. The conductivity results and mechanisms have rewritten. Please see the revised manuscript (Line 330-336, p11-12), and Figure 10.
Point 7: The sentence “There are two conductive mechanisms existing in the microsphere filled polymer: DC conductive (Ni@PS) and capacitive (polymer)conductive” is not correct. The author should refer to the to the universal law of Jonsher. The DC conductivity is the independent frequency conductivity at low values of frequency. And the DC conductivity occurs in polymer such as epoxy resin.
Response 7: Revised accordingly.
Point 8: Line 294 change “captative” by “capacitive”.
Response 8: Done accordingly. Please see the Response 6.
Point 9: The value given for the percolation threshold value are not correct. The author should use the percolation law to determine the value.
Response 9: Done accordingly. Please see the Response 6.
Point 10: To go deeper in the interpretation, other dielectric measurements at low frequency (0.1 Hz) and as a function of the temperature would be helpful.
Response 10: Thank you for professional suggestion. Because the limited frequency range of the instrument applied is between 20 Hz and 1 MHz in experiment, we could not carry out the dielectric measurements at low frequency (0.1 Hz) currently. However, we will do our best to do it according to the suggestion in the coming work.
Reviewer 2 Report
The research «AC electric-field assistant architecting ordered network of Ni@PS microspheres in epoxy resin to enhance conductivity» is devoted interesting topic. The work presents numerous informative micrographs of the obtained materials. Preparation of Ni@PS microspheres and Ni@PS/EP composites is described in detail. The results of the study by various methods are presented. The introduction substantiates the relevance of the research direction. The conclusion summarizes the main results and shows the main numerical parameters that confirm the significance of the study. Improvement of the manuscript can be achieved by paying attention to the following points:
1. The article does not provide sufficient explanation for the choice of 1000 Hz frequency for analysis of electrical conductivity of Ni@PS/EP composites with different Ni@PS content (Fig.10c).
2. The authors have a sufficient amount of experimental data to calculate the percolation threshold (see Figure 10 c), but this has not been done. The discussion on lines 316-325 is inconclusive. Need to add a calculation of the percolation threshold value, for example, as Journal of Applied Polymer Science, 51168.
3. In line 322, the authors refer to 38 - Untereker’s report, but in the links this work is named serial number 39. It is necessary to check the correctness of the references in the article.
4. FTIR data is not well described.
Reconsider after major revision.
Author Response
Response to Reviewer 2 Comments
Comments and Suggestions for Authors
The research «AC electric-field assistant architecting ordered network of Ni@PS microspheres in epoxy resin to enhance conductivity» is devoted interesting topic. The work presents numerous informative micrographs of the obtained materials. Preparation of Ni@PS microspheres and Ni@PS/EP composites is described in detail. The results of the study by various methods are presented. The introduction substantiates the relevance of the research direction. The conclusion summarizes the main results and shows the main numerical parameters that confirm the significance of the study. Improvement of the manuscript can be achieved by paying attention to the following points:
Point 1: The article does not provide sufficient explanation for the choice of 1000 Hz frequency for analysis of electrical conductivity of Ni@PS/EP composites with different Ni@PS content (Fig.10c).
Response 1:Corrected accordingly. The conductivity results and mechanisms have rewritten. Please see the revised manuscript (Line 332-336, p12), as shown as following:
It can be noted from Figure 10 that the conductivity of all the samples obtained at lower (< 1000 Hz) is frequency-independence. To investigate the effect of alignment of Ni@PS on the conductivity without considering the contribution of the frequency. The values of electrical conductivity of the samples with different Ni@PS content obtained at frequency of 1000 Hz were plotted, as shown in Figure10 c.
Point 2: The authors have enough experimental data to calculate the percolation threshold (see Figure 10 c), but this has not been done. The discussion on lines 316-325 is inconclusive. Need to add a calculation of the percolation threshold value, for example, as Journal of Applied Polymer Science, 51168.
Response 2:we have calculated the percolation threshold according to the suggestion, please see the revised Figure 10 d.
The discussion on lines 316-325 has been rewritten, as shown in the revised manuscript line 348-358, p12. In addition, the following paper has been read to help us calculate percolation threshold, and cited as 44;
- Vikulova, M.; Tsyganov, A.; Bainyashev A, Artyukhov D.; Gorokhovsky, A.; Muratov, D.; Gorshkov, N. Dielectric properties of PMMA / KCTO (H) composites for electronics components. J. Appl. Polym. Sci. 2021, 138. 51168. 10.1002/app.51168.
Point 3: In line 322, the authors refer to 38 - Untereker’s report, but in the links this work is named serial number 39. It is necessary to check the correctness of the references in the article.
Response 3:We have checked all the references. The references 38 has been changed into 45 and marked in the manuscript and reference’s part.
Point 4: FTIR data is not well described.
Response 4:FTIR data has been rewritten, as shown in the following (line 177-180, p5: In the original PS, the absorption peak at 1450~1600 cm-1 is due to the aromatic ring and the peaks at 700~755 cm-1 are from vibration of -CH2 of PS, while in the FTIR of Ni@PS microspheres, the absorptions for PS have disappeared, indicating the PS has been covered by the Ni.
Round 2
Reviewer 1 Report
The manuscript has been sufficiently improved to warrant publication in Polymers. The author should only corrected the following part concerning the conductivity.
"In the figure 10, plateau of nearly frequency-independent conductivity is referred to as the DC conductivity. The curve of frequency-dependent area beyond this plateau is referred to as AC conductivity. At a given temperature, the response of AC conductivity
with frequency could be expressed by the empirical power law, that first described by Jonscher [38], as shown below:
???(?) = ??? +?(2??)?
, where f is the frequency , A and n are parameters which are dependent upon temperature, morphology, movement of polymer chains, and composition. "
Author Response
We appreciate reviewer's suggestion. We have corrected accordingly. as shown in the belowing:
In the figure 10, plateau of nearly frequency-independent conductivity is referred to as the DC conductivity. At a given temperature, the response of conductivity with frequency could be expressed by the empirical power law, that first described by Jonscher [38], as shown below: ???(?)=??? +?(2??)?, where f is the frequency , A and n are parameters dependent temperature, morphology, movement of polymer chains, and composition.
Reviewer 2 Report
My comments are corrected. I recommend to accept in present form.
Author Response
Thank the reviewer for your professinal suggestion and hard work on our manuscript.